# Structural insights into mechanisms of zinc scavenging by the *Candida albicans* zincophore Pra1

Alexandre Nore[1], Elena Roselletti[1], Tanmoy Chakraborty[1], Nicha Särkkä[2], Rajika L. Perera [3,5], Duncan Wilson [1] ✉ & Johanna L. Syrjänen[2,4] ✉

*Candida albicans* causes over 400,000 life-threatening, and an additional half a billion of mucosal infections annually. In response to infection, the host limits essential micronutrient availability, including zinc, to restrict growth of the invading pathogen. As assimilation of zinc is essential for *C. albicans* pathogenicity, limitation induces secretion of the zincophore protein Pra1 to scavenge zinc from the host. Pra1 also plays a number of important roles in host-pathogen interactions and is conserved in most fungi. However, the structure of fungal zincophores is unknown. Here, we present cryo-EM structures of *C. albicans* Pra1 in apo- and zinc-bound states, at 2.8 and 2.5 Å resolution respectively. Our work reveals a hexameric ring with multiple zinc binding sites. Through genetic studies, we show that these sites are essential for *C. albicans* growth under zinc restriction but do not affect the inflammatory properties of Pra1. These data create a foundation for future work to explore the structural basis of Pra1-mediated host-pathogen interactions, *C. albicans* zinc uptake, as well as therapeutics development.

*C andida albicans* is one of the most common fungi in humans, typically living commensally within the oral cavity, gastrointestinal, and urogenital tracts[1]. However, overgrowth of *C. albicans* can cause infections, which in otherwise healthy individuals commonly present as oral or vaginal thrush (candidiasis) and in the immunocompromised can lead to life-threatening invasive candidiasis of internal organs or bloodstream infection (candidemia)[2]. *C. albicans* is responsible for over half a billion mucosal infections and 400,000 life-threatening invasive infections every year and, as such, has recently been recognized by the World Health Organization as a critical priority pathogen[3].

The ability of *C. albicans* to capture zinc from its environment is critical to its pathogenicity[4]. During infection, the host employs a strategy called nutritional immunity to restrict the availability of essential trace elements to the invading pathogen[5]. For example, inflamed tissue can contain high levels of the host zinc-chelating protein calprotectin[6]. In response to such conditions of low zinc availability, *C. albicans* secretes a zinc-scavenging protein called Pra1 to capture zinc from the host[4]. Proteinaceous metallophore systems such as that mediated by Pra1 are highly unusual; small molecule metallophores, such as siderophores in the case of iron ion capture, are more typical[7]. This proteinaceous 'zincophore' is entirely unique to the fungal kingdom and arose in an ancient lineage approximately 500 million years ago. Following extracellular secretion, Pra1 associates with the fungal cell via a cognate receptor called Zrt101; the *PRA1* and *ZRT101* genes are encoded at the same genetic locus[4].

As well as capturing zinc for the fungus, Pra1 can influence immune function at multiple levels. Pra1 interacts with several components of the mammalian complement pathway, including factor H, C3, and C4b[8–10]. The protein also influences fungal recognition by immune phagocytes and serves as a ligand for the neutrophil integrin

[1]Faculty of Health and Life Sciences, Medical Research Council Center for Medical Mycology, Geoffrey Pope Building, University of Exeter, Exeter, UK. [2]Institute of Biotechnology, HiLIFE, University of Helsinki, Helsinki, Finland. [3]Poseidon Laboratory, Pasadena, CA, USA. [4]W.M Keck Structural Biology Laboratory, Cold Spring Harbor Laboratory, Cold Spring Harbor, New York, NY, USA. [5]Present address: Ophidion Inc, Pasadena, CA, USA. ✉e-mail: duncan.wilson@exeter.ac.uk; johanna.syrjanen@helsinki.fi

Mac-1[11]. We have recently shown that Pra1 is responsible for the inflammatory immunopathology of vulvovaginal candidiasis[12]. Although the physiochemical properties of individual Pra1 peptide fragments have been investigated, the mechanisms of Pra1 zinc sequestration remain unexplored on a structural level[7,13]. In this study, we present cryo-EM structures of *C. albicans* Pra1 in the apo-state as well as in the zinc-bound state at 2.8 Å and 2.5 Å resolution, respectively. Our study reveals a hexameric, wheel-like structure, with key zinc-binding sites on the exterior rim of the wheel. Further, we show that *C. albicans* carrying mutations of these sites shows severely impaired growth under low zinc conditions, but this does not affect Pra1's immuno-stimulatory properties. Together, our structural and genetic discoveries suggest possible molecular mechanisms for zinc capture by Pra1. Our work is foundational for further structural, biochemical and genetic studies unraveling the molecular steps in the zincophore activity of Pra1. Because of the central role that Pra1 plays in *C. albicans* pathogenicity, it is a key target for the development of new therapeutics. These structures are a substantial step forward in aiding these efforts.

## Results

We determined the structure of recombinantly expressed *C. albicans* Pra1 to 2.5 Å resolution (Supplementary Fig. 1 and Supplementary Fig. 2). The structure reveals a hexameric ring-like assembly (Fig. 1a) in which each subunit is related to its neighbor by a 180° rotation (Fig. 1b). This leads to two distinct inter-subunit interfaces (Fig. 1b–f). From examination of these interfaces, we suggest that the hexameric assembly formed by Pra1 subunits comprises a trimer of dimers. To form a dimer, the N-terminus of a subunit hugs across the main body of its neighbor like an outstretched arm or brace (Fig. 1b–d). As illustrated in Fig. 1c, these interactions are primarily mediated by salt bridges (Arg 34 with Asp 71; Asp 36 with Arg 75; Arg 75 with Asp 107) and hydrogen bonding (Tyr 31 with Arg 75; Trp 33 with Asp 230; Trp 37 with the backbone oxygen of Ala 106) of the N-terminal brace with its neighboring subunit. In addition to the extensive N-terminal interactions with the neighboring subunit, the polar interaction of Ser 93 with Gln 188 contributes to the intra-dimer interactions (Fig. 1d). In contrast to the intra-dimer interactions, the inter-dimer interactions are almost solely driven by polar interactions that involve residues Thr 50, Gln 54, Thr 57, Tyr 152, Gln 155, Thr 154, and Ser 159 (Fig. 1e, f).

Pra1 is 299 amino acids in length (including its N-terminal signal peptide), and our cryo-EM structure allows the visualization of residues 31–251, as well as some post-translational modifications. In agreement with original observations of *C. albicans* fungal culture, Pra1 is extensively glycosylated[14]. Asn 48, Asn 89, Asn 135, and Asn 208 all show evidence of N-linked glycosylation. The glycosylated residues are all located in close proximity to subunit interfaces (Fig. 2a and Supplementary Fig. 3). Asn 48 is close to the inter-dimer interface (Supplementary Fig. 3a). In contrast, Asn 89, Asn 208, and Asn 135 are all located close to the intra-dimer interface, leading to a total of six glycosylation sites in this area (Supplementary Fig. 3b).

Examination of a single subunit shows that the N- and C-termini are located on opposite faces of the protomer (Fig. 2a). The 180° rotational relationship between neighboring subunits means that each of the "top" and the "bottom" faces of the hexameric ring has three N-termini and C-termini. The core of the protomer is made up of five short helices, one longer helix and one beta-sheet (Fig. 2a). The longest helix (residues 52–76) spans 24 amino acids across the length of the subunit and faces the interior of the hexameric ring (Fig. 2a). The beta-sheet, consisting of three beta-strands (within residues 112–142), is on the exterior of the hexameric ring (Fig. 2a). Each Pra1 subunit also has three intramolecular disulfide bonds, all in close proximity to the inter-dimer interface (Fig. 2a). Indeed, the overall fold of a Pra1 protomer is similar to that of HEXXH + D type metalloproteases (Supplementary

Fig. 4a; a structural superposition of Pra1 with deuterolysin from *Aspergillus oryzae*; PDB code 1EB6, yields an RMSD = 3.74 Å).

This raises the intriguing question of what the Zn$^{2+}$ binding site in Pra1 looks like and how it compares to that of the catalytically active metalloproteases. From our cryo-EM structure of Pra1 prepared in the presence of 1 mM ZnSO$_4$, we observe that the cryo-EM map is continuous amongst a cluster of three histidine residues (His 178, His 182 and His 193), which all point towards each other, consistent with the coordination of a metal ion (Fig. 2b). In this subunit (subunit B), the center of the zinc ion is equidistant to the center of the N3-nitrogen in the imidazole ring of each histidine residue. This distance is 2.3 Å. There is an extension of cryo-EM density in the metal ion binding site, likely indicating the presence of a water molecule to complete the tetrahedral coordination of the zinc ion. Although we generated two final maps for the Pra1:Zn$^{2+}$ structure, one with c1 (no symmetry) and the other with three-fold c3 symmetry imposed, we analyzed and deposited the c1 map. In the c1 map, we are able to observe that the distances from the center of the zinc ion to the center of the N3-nitrogen in the histidine residue vary by subunit (Supplementary Fig. 5). For instance, in subunit A, the distance from the center of the zinc ion to the center of the N3-nitrogen of His 193 and His 182 is 2.3 Å. In contrast, the distance from the center of the zinc ion to the center of the N3-nitrogen of His 178 is 2.7 Å. These variations may be indicative of differential binding modes and may be important in the molecular mechanism of zinc ion binding by the Pra1 hexamer.

We observe that the location of metal ion binding in Pra1 is conserved with that of the well characterized acid metalloprotease deuterolysin, by overlaying the Pra1 protomer with that of *Aspergillus oryzae* deuterolysin (Supplementary Fig. 4b). While His 182 and His 178 in *C.albicans* Pra1 are conserved as His 132 and His 128 in *A.oryzae* deuterolysin, the canonical catalytically active glutamic acid and aspartic acids characteristic of metalloproteases, Glu 143 and Asp 129 in *A.oryzae* deuterolysin, are not conserved in Pra1. In contrast, Pra1 harbors His 193 in the *A.oryzae* deuterolysin Glu 143 position and Arg 179 in the *A.oryzae* deuterolysin Asp 129 position (Supplementary Fig. 4b). In contrast to deuterolysin, this metal ion coordination site in Pra1 thus binds to Zn$^{2+}$ but is unlikely to be involved in proteolysis. In the context of the overall Pra1 architecture, these six Zn$^{2+}$ binding sites are located on the outer face of the hexameric ring (Fig. 2c).

To further compare the Zn$^{2+}$-bound structure to Pra1 in the apo-state, we collected cryo-EM data on *C. albicans* Pra1 in MES-NaOH pH 6.0, thereby ensuring protonation of histidine residues and release of any zinc ions due to electrostatic repulsion (Supplementary Fig. 6). As for the Pra1:Zn$^{2+}$ complex, we determined both c1 and c3 maps but used the c1 map for analysis in this study. A comparison between the maps of Pra1 in complex with Zn$^{2+}$ and Pra1 in MES pH 6.0 at the same contour level suggests that the His 178, His 182 and His 193 triad no longer coordinate the metal ion under low pH conditions in the absence of zinc (Fig. 2d). The lack of metal ion coordination, and the change in pH, does not lead to global conformational changes in the resolved portions of Pra1. Indeed, a structural superposition of the hexamers in the presence and absence of Zn$^{2+}$, shown in Supplementary Fig. 7, has an RMSD = 0.175 Å.

Our zincophore structure reveals an important role for the histidine residues at positions 178, 182, and 193 in zinc ion coordination. We next investigated the evolutionary and functional implications of these findings. The *PRA1* gene is ancient, having originated in an early fungal lineage, and is present in most extant species[4,15]. The histidine triad contains an evolutionarily conserved motif of HRXXH + H (the histidine residues of which correspond to His 178, His 182, and His 193, respectively, in *C.albicans*). *C. albicans* Pra1 also contains an HAXXH motif (residues 68–72) and a C-terminal CHTHXXGXXHC motif (residues 289–299). We did not observe structural changes or zinc ion binding at pH 6.0 or pH 8.0 within proximity to residues 68–72. Whilst a short peptide spanning residues 289–299 does bind zinc ions, it is

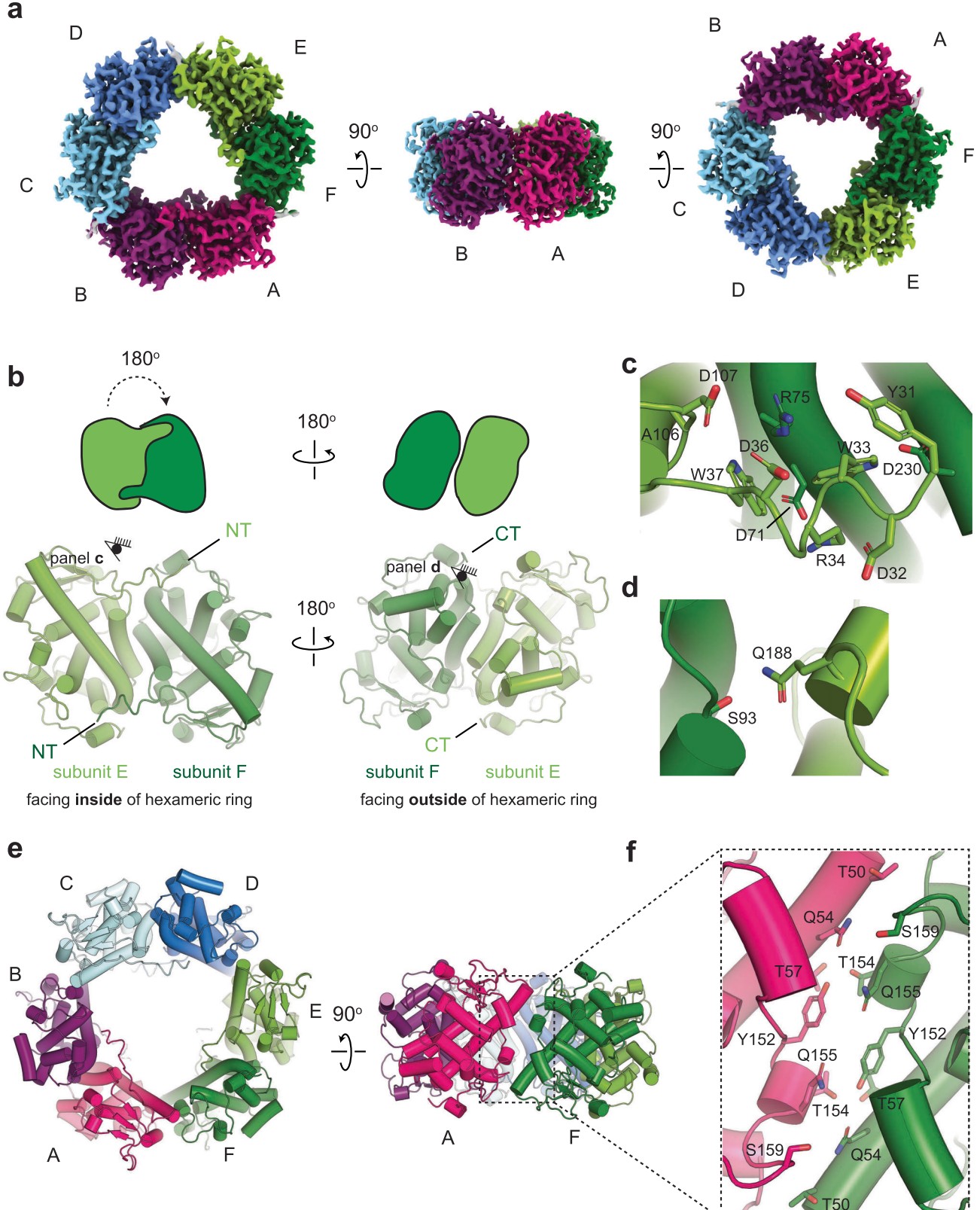

**Fig. 1 | The cryo-EM structure of Pra1 reveals a hexameric assembly. a** The cryo-EM map of *C. albicans* Pra1. **b** The hexamer consists of an assembly of a trimer of dimers. A dimer is shown in split pea green (subunit E) and forest green (subunit F). Each Pra1 subunit is related to its neighbor by a 180° rotation, resulting in two distinct interaction interfaces per subunit. Subunits E + F, A + B, and C + D make up the dimers. **c** The N-terminus of a Pra1 subunit forms a brace or arm that is held across part of its neighboring subunit to form a dimer. Here, the N-terminus of subunit E is shown forming a brace across part of subunit F. The N-terminal brace intra-dimer interactions are primarily mediated by salt bridges and hydrogen bonds. **d** Apart from the N-terminal brace interactions, Q188 and S93 form polar interactions across the intra-dimer interface. **e, f** The interactions at the inter-dimer interface are polar in nature.

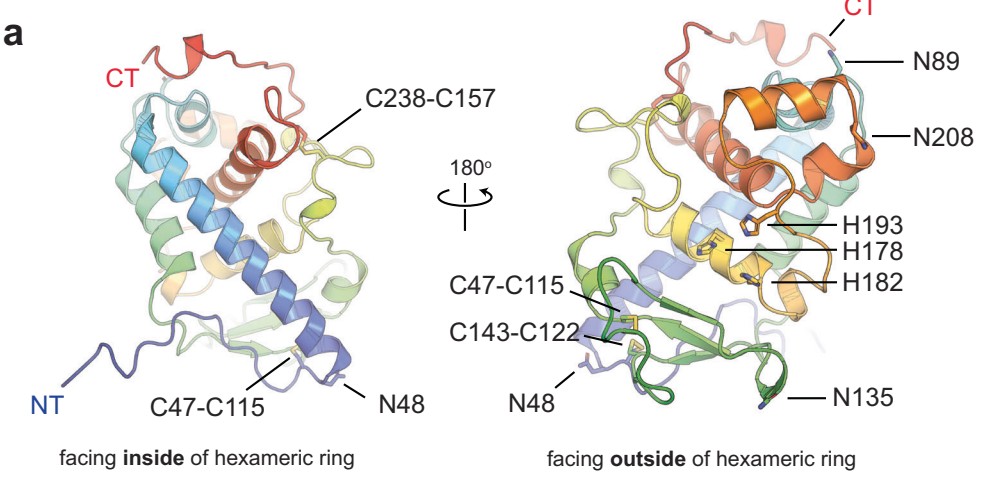

**Fig. 2 | A triad of histidine residues forms a zinc ion binding site on a Pra1 subunit on the exterior of the hexameric ring. a** The N-terminal and C-terminal ends of the protein are on opposite sides of Pra1. A single subunit consists of several short helices, as well as a long helix (24 amino acids in length), and a small beta-sheet. In addition, a Pra1 subunit harbors three disulfide bridges and four N-linked glycosylation sites. **b** A close-up of the histidine triad comprised of His 178, His 182, and His 193 in the presence of 1 mM ZnSO₄ reveals cryo-EM map density consistent with a metal ion coordinated by the triad. In this subunit (B), the distance from the center of the zinc ion to the center of the N3-nitrogen in the imidazole ring of the histidine residue is 2.3 Å for all three histidine sidechains. **c** The locations of the histidine triad consisting of His 178, His 182, and His 193 on the exterior ring are indicated in magenta space fill. Each subunit harbors one histidine triad, leading to six zinc ion binding sites on the exterior of the Pra1 ring. **d** A close-up of the histidine triad in MES pH 6.0 reveals a lack of metal ion binding under these conditions, as indicated by the absence of continuous cryo-EM density between the three histidines, in contrast to panel (**b**).

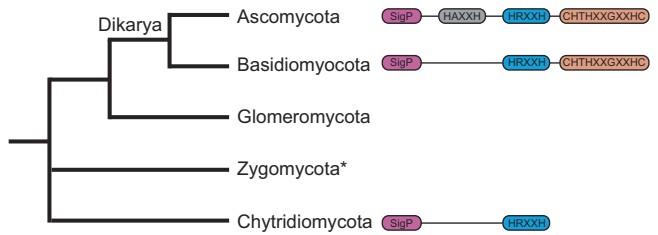

**Fig. 3 | Simplified overview of fungal phylogeny and the most common Pra1 signal peptide and predicted zinc-binding motifs in each phylum.** The asterisk denotes that Zygomycota is now two phyla, the Mucoromycota and Zoopagomycota.

located at the end of what is predicted by AlphaFold to be a long, unstructured C-terminus of the Pra1 protomer (Supplementary Fig. 8) and is not resolved in our cryo-EM structures[7].

BLASTp analysis of the *C. albicans* Pra1 primary amino acid sequence at NCBI identified hits only in fungi and in a single archaeon. Figure 3 shows a simplified overview of the phylogenetic relationship amongst fungal phyla. We observed 1336 hits in ascomycete fungi, 150 in basidiomycetes, two in Chytridiomycetes, and two in a Bathyarchaeota archaeon. Reciprocal BLASTp of the archaeal sequences identified orthologues in ascomycete fungi, indicating possible horizontal transfer of the gene from fungi to this recently described archaeon.

Phylogenetic analysis of fungal Pra1 orthologs revealed the following. Most ascomycete fungi have orthologues with similar histidine-rich motifs as *C. albicans* Pra1. Several ascomycete fungi, including the medically relevant *Blastomyces percusus*, have lost the C-terminal CHTHXXGXXHC motif (Supplementary Fig. 9). In basidiomycetes, the C-terminal motif has been retained, whereas none of the analyzed sequences from basidiomycetes possessed the HAXXH motif. Finally, the two chytrid proteins each possess the core HRXXH motif and lack both HAXXH and CHTHXXGXXHC. As we have reported previously, multiple fungal species have lost the Pra1 encoding gene[4].

Because chytrids are an early diverging fungal phylum, it would appear that the origin of the Pra1 zincophore gene actually lies in a very early fungal lineage, probably upon mutation of the catalytic glutamic acid residue of a metalloprotease to arginine (Supplementary Fig. 4). Following the divergence of the Dikarya from basal fungal lineages, the contemporary Pra1 gained its CHTHXXGXXHC motif—incidentally, the encoding gene formed a syntenic relationship with its receptor (Zrt101) in the Dikarya[4]. Finally, following divergence of the ascomycetes and basidiomycetes, the ascomycetes gained an additional HAXXH motif, and a small number of species lost the C-terminal CHTHXXGXXHC motif.

From this analysis, together with our structural observations (Fig. 2b), we conclude that the 178-HRXXH-182 motif, together with His 193, represents the core zincophore domain of Pra1. We therefore created a modified *C. albicans* strain that expresses Pra1, but with disruption of the histidine triad (H178A and H182A), and compared this strain to *C. albicans* wild type (Wt), *PRA1* gene deletion mutant (*pra1Δ*), and genetic revertant (*pra1Δ+PRA1*).

When cultured in acidic (pH 5) minimal media without zinc restriction, all *C. albicans* strains grew equally well (Fig. 4a). Mild zinc depletion, elicited with 0.5 mM EDTA, delayed the growth of all strains, whilst strong zinc depletion (2 mM EDTA) effectively prevented growth (Fig. 4b, c). In neutral/alkaline media without zinc restriction, all strains again grew equally well, albeit slightly slower than under acidic conditions (Fig. 4d). This is because yeast-fungi generally grow better at acidic pH. Strikingly, upon mild zinc-restriction, wild-type *PRA1*-expressing strains grew, whilst the *pra1Δ* null, and the H178A/H182A Pra1 variant both exhibited severely delayed growth, and upon

strong-zinc depletion, growth of these strains was abolished (Fig. 4e, f). From these growth assays together with their respective calculated areas (Area Under Curve, Fig. 4g), it would appear that *C. albicans* specifically utilizes the Pra1 histidine triad for growth under zinc limitation at neutral-alkaline pH. We recombinantly expressed and purified the Pra1 Histidine-to-Alanine Pra1 H178A/H182A mutant and compared its size-exclusion chromatography (SEC) profile to that of wild-type Pra1 in order to verify that the effects which we observe on growth are due to the point mutations themselves within the context of a properly assembled protein. As can be seen in Supplementary Fig. 10, the behavior of recombinant Pra1 H178A/H182A as evaluated by SEC retention volume and peak shape is very similar to that of wild-type Pra1. This indicates that the effect on cell growth is driven by these specific residues rather than a global destabilization of the H178A/H182A Pra1 protein.

Besides capturing zinc, we have recently shown that Pra1 is a major driver of the inflammatory immunopathology of vulvovaginal candidiasis[12]. However, it is not yet known whether these two functions are related. Although we did not detect zincophore activity of the HAXXH motif (residues 68–72) in our structural analysis, this site has been implicated in $Zn^{2+}$-binding, and we reasoned that in a more complex host-pathogen interaction setting, it could also influence immune recognition[7]. Therefore, for these experiments, we created a second modified Pra1 in which the H178A/H182A histidines of the histidine triad, as well as the 68-HARDH-72 motif, were substituted for alanine.

To test whether zincophore function is important for neutrophil recognition, we incubated our different *C. albicans* strains in RPMI tissue culture media for five days. Under these culture conditions, Pra1 is dispensable for growth, and all strains grow to similar levels[12]. The fungal culture filtrates were added to the lower compartments of a chemotaxis assay plate; freshly isolated, labeled human neutrophils were then added to the upper compartment and allowed to migrate for 2 h. As we have observed previously, *C. albicans* wild-type supernatant drove robust neutrophil recruitment, and this was blocked by deletion of *PRA1*. Genetic complementation of the *pra1Δ* mutant with a wild-type copy of *PRA1* restored neutrophil recognition. Interestingly, genetic complementation of *pra1Δ* with the modified Pra1 variant fully restored neutrophil recognition to wild-type levels (Fig. 5). Therefore, the histidine triad is essential for *C. albicans* growth under zinc limitation, but is not involved in neutrophil recognition.

## Discussion

Infections caused by *C. albicans* trigger a nutritional immune response by the host to inhibit pathogen growth. This results in environments where micronutrients such as zinc are scarce. To sequester zinc ions, *C. albicans* deploys its zinc-scavenging machinery, a key component of which is the zincophore protein Pra1.

In this study, we determined cryo-EM structures of the *C. albicans* Pra1 both in the presence and absence of zinc, revealing a hexameric ring-like assembly. The 178-HRXXH-182 motif was previously described as a predicted zinc ion binding site[4]. Our study revealed a conserved histidine triad comprising His178, His182, and His193. Our Pra1 structure determined in the presence of zinc showed cryo-EM density within this triad consistent with zinc ion coordination. Each Pra1 protomer hosts one triad, leading to six triads on the outside rim of the hexameric Pra1 assembly. We postulated that these triads are important for zinc sequestration and hence needed for *C. albicans* growth under zinc-limited conditions. We made a variant Pra1-His178Ala/His182Ala strain to disrupt the integrity of the histidine triad and subsequently showed that this strain indeed has severe growth defects under low zinc conditions, specifically in a neutral-alkaline pH environment.

We have recently shown that the Pra1 protein triggers the host inflammatory response during vaginal candidiasis, the most common

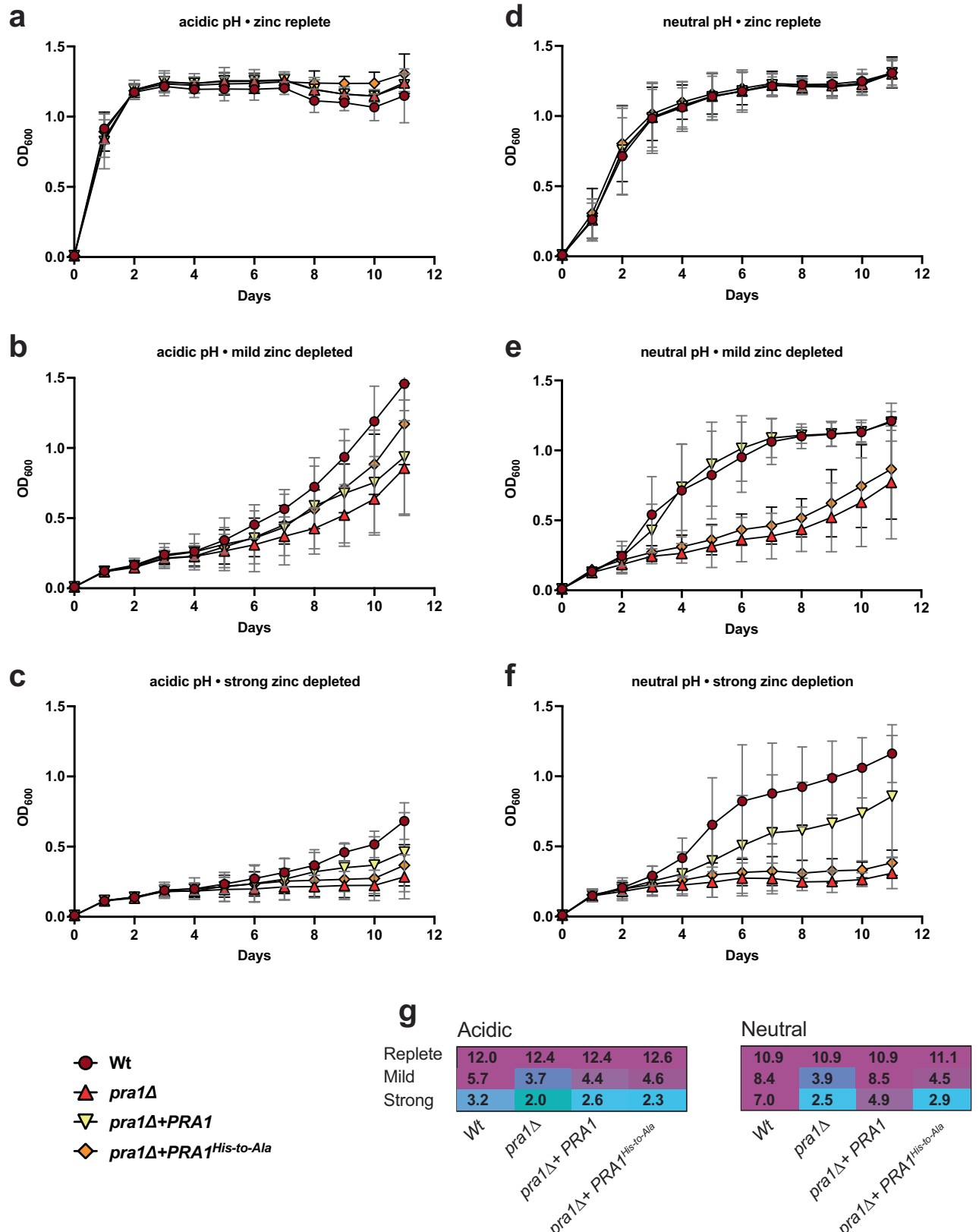

**Fig. 4 | The HRFWH motif is essential for *Candida albicans* growth under zinc restriction at neutral pH. a–f** Wild type (Wt), *PRA1* deletion mutant (*pra1Δ*), *pra1Δ* expressing wild type *PRA1* (*pra1Δ+PRA1*) and *pra1Δ* expressing 178-HRFWH-182 His-to-Ala variant (*pra1Δ+PRA1^His-to-Ala^*) were cultured in acidic (unbuffered, starting pH ~5) or neutral/alkaline (buffered to pH 7.4) zinc-free minimal medium (YNB-zinc-dropout; 0.5% glucose) supplemented with 1 µM zinc. To elicit mild- and strong-zinc depletion, media were supplemented with 0.5 mM or 2 mM EDTA, respectively. Cells were cultured at 30 °C and 180 rpm. Growth was assessed by measuring $OD_{600}$

for 11 days (*n* = 8). Results are the mean of four independent experiments. The data points represent the mean, and the error bars represent the standard deviation. Dark red circles represent Wt, red triangles represent *pra1Δ*, inverted yellow triangles represent (*pra1Δ+PRA1*), and orange diamonds represent (*pra1Δ+PRA1^His-to-Ala^*) strains. **g** The area under the curve was calculated for each growth curve and plotted with magenta coloring for high and cyan coloring for low measurements. Source data are provided as a Source Data file.

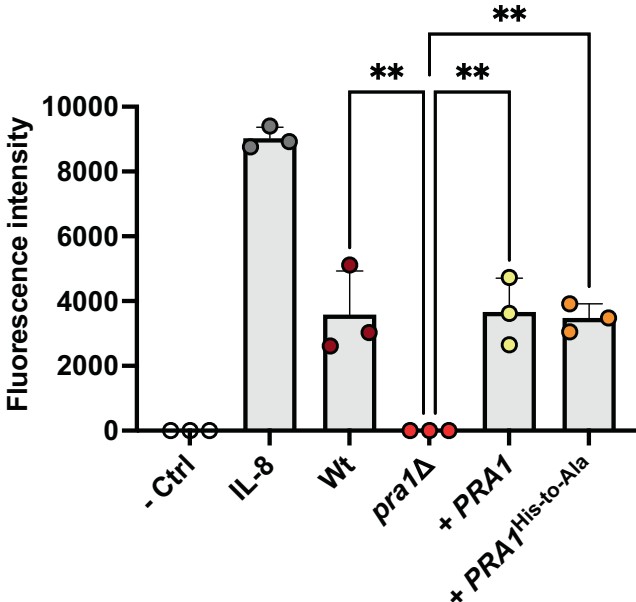

**Fig. 5 | Zinc-binding motifs of Pra1 are not required for neutrophil recognition.** Wild type (Wt), *PRA1* deletion mutant (*pra1Δ*), *pra1Δ* expressing wild type *PRA1* (*pra1Δ +PRA1*), and *pra1Δ* expressing 178-HRFWH-182/68-HARDH-72 His-to-Ala variant (*pra1Δ +PRA1^His-to-Ala*) were incubated in RPMI for five days. Culture filtrate, unconditioned media (RPMI), or media containing the neutrophil chemotactic factor, IL-8 (100 ng/ml), were added to the lower compartment of the chemotaxis plate. Freshly isolated human neutrophils were fluorescently labeled with Calcein and added to the upper compartment. Following 2 h incubation, neutrophil chemotaxis was determined by measuring the fluorescence intensity (at 485/530 nm) in the lower compartment (*n* = 3). The bars represent the mean with standard deviation. **$p < 0.01$. *p*-values: *pra1Δ* vs. Wt, $p = 0.0027$; *pra1Δ* vs. + PRA1, $p = 0.0023$; *pra1Δ* vs. + PRA1His-to-Ala, $p = 0.0032$. The statistical test is a one-way ANOVA with Dunnett's post-test amongst *C. albicans* culture filtrate samples. The chemotaxis assays were performed against supernatants from three independent fungal cultures per strain using neutrophils isolated from three independent donors. Source data are provided as a Source Data file.

debilitating fungal infection in the world[12]. Critically, our current study shows that the histidine triad, whilst crucial for zincophore activity, is entirely dispensable for neutrophil recognition and is thus unlikely to contribute to the inflammatory response by the host. This is important because it means that future interventions may be tailored to target specific properties of Pra1, the structural frameworks for which are provided by our cryo-EM data.

Our phylogenetic analysis of the Pra1 gene revealed its origin in a very early fungal lineage. The zincophore activity of Pra1 orthologs is likely also conserved. For instance, the *Aspergillus fumigatus* orthologue (AspF2) is required for growth under zinc limitation and has recently been shown to have zinc-binding properties[13,16]. Moreover, amino acid sequence alignment between *C. albicans* Pra1 and *A. fumigatus* AspF2 reveals that the histidine triad, which we identified in Pra1, is conserved in AspF2.

A key question arising from our study is that of the specific mechanism by which zinc ion coordination by the histidine triad contributes to zinc scavenging by Pra1. The C-terminal residues, which harbor conserved histidine-rich regions, are not resolved in our Pra1 cryo-EM maps. This is most likely due to the inherent conformational flexibility of the linker sequences that join the structured core of the protein to the C-terminal histidine-rich regions. The relationship, if any, between these C-terminal zinc-binding sites and the histidine triads on the outer rim of the Pra1 assembly remains to be defined. There are two possible mechanisms of Pra1-mediated zinc ion capture consistent with our data. One possibility is that the C-terminal regions function like the arms of an octopus, foraging for zinc ions, which are then transferred to the triads on the outside rim of Pra1. In this model, the histidine triads act as zinc ion storage units (Model 1 in Fig. 6). Alternatively, the histidine triads may directly capture zinc ions (Model 2 in Fig. 6). In both scenarios, our data suggest that the Pra1 histidine triads must be coordinating $Zn^{2+}$ in order for *C. albicans* to grow under zinc-limiting conditions. This further suggests that histidine triad zinc ion coordination is essential either in the zinc ion capture step by Pra1, or in the transfer of zinc ions from Pra1 to the zinc transporter Zrt101, or both. The work described here is thus foundational for unraveling

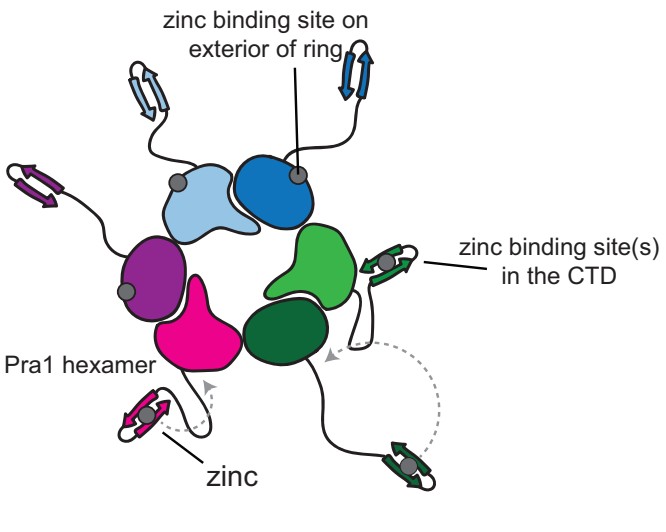

**Model 1**

The C-terminal arms capture zinc and transfer them for storage to the histidine triads on the outer rim of Pra1

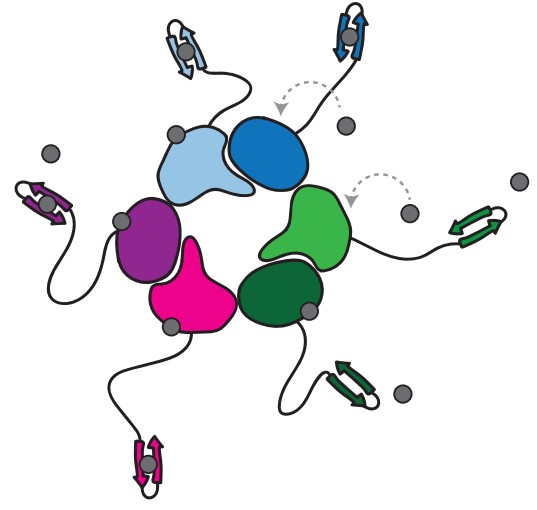

**Model 2**

The histidine triads directly participate in the capture of zinc

**Fig. 6 | Possible molecular models of zinc ion capture by Pra1.** Model 1 posits that the histidine triads function as zinc ion storage units following capture of zinc ions by the C-terminal arms (left). In Model 2, zinc ions are sequestered directly by the histidine triads on the outer rim (right).

the structural basis of the Pra1-Zrt101 zinc uptake mechanism in future work as well as for the development of therapeutics that target Pra1 in *C. albicans* infections.

## Methods

### Cloning, expression, and purification of Pra1 and the Pra1 H178A/H182A mutant

The codon-optimized full-length *PRA1* (excluding the signal peptide) was cloned into the pCDNA3.0 plasmid with an N-terminal His-tag and a TEV protease cleavage site, using standard cloning techniques. Mutations to create H178A/H182A Pra1 were introduced using overlap PCR with codon-optimized full-length *PRA1* as a template. This insert was cloned into the pCDNA3.0 plasmid in a similar manner to full-length Pra1. Expi293 cells (50 ml culture) were transfected with the construct according to the manufacturer's instructions, and the supernatant was collected five days post-transfection. Protein purification was performed using an ÄKTA Go system. The supernatant was loaded onto a 1 ml HisTrap Excel column pre-equilibrated with 10 column volumes (CV) of 50 mM Tris-HCl (pH 8.0), 500 mM NaCl. After washing with 10 CV of the same buffer, the protein was eluted with 5 CV of 50 mM Tris-HCl (pH 8.0), 500 mM NaCl, and 400 mM imidazole. Eluted fractions were pooled and buffer-exchanged into TEV cleavage buffer (50 mM Tris-HCl, pH 8.0, 0.5 mM EDTA, 1 mM DTT). TEV cleavage was performed overnight using a 1:1000 molar ratio of TEV protease to target protein. The following day, the digestion mixture was passed over 0.5 ml Ni-NTA agarose resin (Thermo Scientific) using gravity flow to remove His-tagged TEV and other contaminants. The flow-through containing the cleaved protein was concentrated and subjected to size-exclusion chromatography (SEC) on a Superose 6 column in 50 mM Tris-HCl, pH 8.0, 500 mM NaCl, 10% glycerol. Peak fractions were collected, analyzed via SDS-PAGE, and pooled for further concentration and preparation for cryo-EM. Analytical SEC of Pra1 and H178A/H182A Pra1 were performed in 20 mM HEPES-NaOH pH 7.4, 300 mM NaCl, 0.5 mM EDTA buffer using a Superose 6 10/300 Increase column with a flow rate of 0.5 ml/min on an ÄKTA pure system.

### Cryo-EM sample preparation, image collection, and single-particle analysis

Pra1 was extensively buffer-exchanged into either 20 mM MES pH 6.0, 150 mM NaCl or 20 mM Tris-HCl pH 8.0, 150 mM NaCl using spin filter concentrators with a 50 kDa cut-off (Amicon). Prior to sample freezing, BSA (Sigma) was added as an additive at 0.05%. For the Pra1:$Zn^{2+}$ sample, $ZnSO_4$ was added to the sample to a final concentration of 1 mM prior to freezing. Pra1 (3–4 μl) was applied to glow-discharged 1.2/1.3 Quantifoil 300 mesh carbon-coated copper grids or lacey carbon copper grids (Electron Microscopy Services). The grids were blotted for 3.2 s at 85% humidity and 20 °C before plunge freezing into liquid ethane. Datasets were collected using a Titan Krios operated at an acceleration voltage of 300 kV and the GATAN K3 direct electron detector coupled with the GIF quantum energy filter controlled using EPU software. Movies were recorded with a pixel size of 0.861 Å, and a dose rate of 2.1 e/Å²/frame. The program Warp v1.0.9 was used to align movies, estimate the CTF, and pick particles, using a 320 pixel box size[17]. 2D classification and ab initio model non-uniform refinement were performed using cryoSPARC software v4.3.1 and v4.4.0[18]. The data processing workflows for apo-state Pra1 and the Pra1:$Zn^{2+}$ complex are outlined in more detail in Supplementary Figs. 2 and 6, respectively. Local resolutions of the cryo-EM maps were estimated using the program ResMap v.1.0.9[19]. Model building was performed in UCSF Chimera 1.16 and Coot 0.9.8.7[20,21]. Models were depicted in figures using PyMOL 2.5.4 and UCSF ChimeraX 1.7.1[22,23]. The AlphaFold model of *Candida albicans* Pra1 available on the AlphaFold Protein Structure Database was used as a starting model for manual building[24,25]. The models were refined against the cryo-EM maps

## Table 1 | Cryo-EM data collection and refinement statistics

| Data collection and processing | Pra1:$Zn^{2+}$ complex EMD-48872 PDB-9N4D | Pra1 EMD-48869 PDB-9N47 |
|---|---|---|
| Microscope | Titan Krios | Titan Krios |
| Camera | K3/counting | K3/counting |
| Magnification | 105,000 | 105,000 |
| Energy filter | Gatan | Gatan |
| Energy filter slit width (eV) | 20 | 20 |
| Collection software | EPU | EPU |
| Camera | K3/counting | K3/counting |
| Voltage (kV) | 300 | 300 |
| Cumulative exposure (e–/Å²) | 63.2 | 63.2 |
| Exposure rate (e–/Å²/frame) | 2.1 | 2.1 |
| Defocus range (μm) | −1.0 to −2.4 | −1.0 to −2.4 |
| Pixel size (Å) | 0.861 | 0.861 |
| Symmetry imposed | C1 (none) | C1 (none) |
| Number of micrographs | 10,785 | 6886 |
| Initial particle images (no.) | 3,486,720 | 1,025,587 |
| Final particle images (no.) | 1,102,286 | 607,405 |
| 0.143 FSC half map masked (Å) | 2.53 | 2.85 |
| 0.143 FSC half map unmasked (Å) | 3 | 3.3 |
| **Refinement** | | |
| Refinement package | Phenix | Phenix |
| Initial model used (PDB code) | The AlphaFold model of *C. albicans* Pra1 as a starting model for manual building | Pra1:zinc model, followed by manual building |
| Model resolution range (Å) | 2.25–3.25 | 2.5–3.5 |
| Model composition | | |
| Non-hydrogen atoms | 10,626 | 10,620 |
| Protein residues | 1326 | 1326 |
| Ligands | ZN:6 | n/a |
| CC map vs. model (%) | 0.90 | 0.90 |
| R.m.s. deviations | | |
| Bond lengths (Å) | 0.006 | 0.004 |
| Bond angles (°) | 1.033 | 0.543 |
| Validation | | |
| MolProbity score | 1.04 | 1.29 |
| Clash score | 0.54 | 1.22 |
| Poor rotamers (%) | 1.84 | 2.11 |
| Ramachandran plot | | |
| Favored (%) | 97.11 | 96.73 |
| Allowed (%) | 2.89 | 3.27 |
| Outliers (%) | 0 | 0 |
| C-beta deviations | 0 | 0 |
| EMRinger Score | 3.39 | 3.19 |
| CaBLAM outliers (%) | 2.23 | 2.61 |

**Table 2 | Strains of *Candida albicans* used in this study**

| Wilson group unique identifier | Name | Genetic status | Source |
|---|---|---|---|
| D2 | Wild type, Wt | BWP17 + CIp30 | PMID: 22438810 |
| D5 | *pra1Δ* | [BWP17] *pra1::HIS1/pra1::ARG4* + CIp10 | PMID: 22761575 |
| D203 | *pra1Δ* Uri- | [BWP17] *pra1::HIS1/pra1::ARG4* | PMID: 22761575 |
| D275 | *pra1Δ+PRA1* | [BWP17] *pra1::HIS1/pra1::ARG4* + CIp10-*PRA1* | This study |
| D282 | *pra1Δ+PRA1* HRFWH His-to-Ala | [BWP17] *pra1::HIS1/pra1::ARG4* + CIp10-*PRA1*-HRFWH His-to-Ala | This study |
| D284 | *pra1Δ+PRA1* HARDH/HRFWH His-to-Ala | [BWP17] *pra1::HIS1/pra1::ARG4* + CIp10-*PRA1*-HARDH/HRFWH His-to-Ala | This study |

in Coot 0.9.8.7 and PHENIX 1.21.2[21,26]. Data collection and refinement statistics are outlined in Table 1. Superpositions of atomic models and subsequent RMSD calculations were performed in PyMOL 2.5.4 using the align algorithm[22].

### Pra1 variant construction

*PRA1* alleles containing 178-HRFWH-182 and 68-HARDH-72/178-HRFWH-182 His-to-Ala coding variants were synthesized (GeneArt), consisting of the upstream intergenic region, open reading frame of allele B of *PRA1,* and 100 base pairs downstream sequence (*Candida* Genome Database)[27]. However, *C. albicans* transformed with these constructs did not express the Pra1 protein. Therefore, versions containing the whole allele (entire upstream and downstream intergenic regions) were generated. To do this, the wild-type *PRA1* was amplified from *C. albicans* genomic DNA using primers (AN-oli56) SalI_pPRA1_F and (AN-oli57) PRA1_UTR554_MluI_R, cloned into TOPO vector, confirmed by Sanger sequencing, and then subcloned into CIp10 for integration[28]. We selected a CIp10-*PRA1* harboring allele B of *PRA1* for subsequent modification. The resulting CIp10-*PRA1* plasmid was digested with *Sal*I and *Nde*I to excise the first 1192 base pairs of insert, containing the upstream intergenic region and the first 684 base pairs of *PRA1* coding sequence. The remaining backbone, 3′ of the gene, and the downstream intergenic region were purified by gel extraction. In parallel, the first 1,192 base pairs of HRFWH and HARDH/HRFWH His-to-Ala coding variants were excised from the synthetic constructs, purified by gel extraction, and independently cloned into the CIp10-*PRA1* backbone fragment. Each (*PRA1* wild type, *PRA1*-178-HRFWH-182 His-to-Ala, and *PRA1*-68-HARDH-72/178-HRFWH-182 His-to-Ala variants) of the plasmids was linearized and integrated at the *RPS1* locus of the *pra1Δ* uridine auxotroph as previously described[4].

### Growth assays

*C. albicans* strains were maintained on YPD agar plates (1% yeast extract, 2% peptone, 2% glucose, 2% agar). A colony was inoculated into SD minimal medium (1X yeast nitrogen base; 2% glucose) and incubated at 30 °C and 180 rpm. Precultures were then washed twice in milliQ water and used to inoculate culture media in 96-well plates to an $OD_{600} = 0.01$. In all cases, the basal media was zinc-free minimal media supplemented with 1 μM zinc (1X YNB zinc-dropout (Formedium); 0.5% glucose; 1 μM $ZnSO_4$). To buffer to neutral/alkaline pH, 80 mM HEPES pH 7.4 was added. To elicit zinc restriction, EDTA was added to a concentration of 0.5 or 2 mM. The complete media, including EDTA, was always first prepared before the addition of *C. albicans* cells. Plates were incubated at 30 °C and 180 rpm, and $OD_{600}$ was measured daily. All strains of *Candida albicans* used in this study are listed in Table 2.

### Neutrophil chemotaxis assay

*C. albicans* strains were cultured for 24 h in SD minimal media at 30 °C, 200 rpm, washed twice in milli-Q water, and resuspended in RPMI without phenol red at $1 \times 10^6$ cells/ml. These were then distributed in triplicate in a 24-well plate, 1 ml per well, and incubated at 24 °C for 5 days, 130 rpm shaking. Conditioned supernatant was collected, filter sterilized (0.2 μm filter, Fisher), and added to the basal compartment of the chemotaxis chamber.

Human peripheral blood neutrophils (PMN), obtained from healthy volunteers, were separated by density gradient centrifugation on FicollPaque™ Plus, followed by the hypotonic lysis of erythrocytes using a 1X red blood cells lysis buffer (10X recipe: $NH_4Cl$ 1.55 M, $NaHCO_3$ 120 mM, EDTA 1 mM, in sterile $ddH_2O$) Using this protocol, we demonstrated the isolation of a neutrophil population with more than 98% purity using flow cytometry[12]. The purity of human neutrophils, before each chemotaxis test, was always checked using a Wright-Giemsa Stain (Abcam) following the kit instructions. The slides were prepared using a Cytospin 4 centrifuge (Thermo Scientific) and then evaluated microscopically using EVOS M5000 at 40x magnification.

Neutrophils were then washed twice with PBS and resuspended in 5 ml RPMI-1640 without phenol red containing 10% heat-inactivated FBS. The cells were then incubated with Calcein AM, a fluorescent cell-permeable derivative of calcein (Sigma Aldrich) (5 μg/ml) for 30 min at 37 °C + 5% $CO_2$. After the incubation, the neutrophils were washed twice with PBS, counted, and resuspended in the same complete medium at $5 \times 10^6$/ml.

Human neutrophil chemotaxis was measured using a 96-well chemotaxis chamber with a 3.2 mm diameter filter with membrane porosity of 5 μm (Neuro Probe). Labeled human neutrophils ($5 \times 10^6$/ml) were transferred into ChemoTx filters placed in the 96-well plates containing 300 μl of either medium alone (RPMI without phenol red, negative control), media containing recombinant IL-8 (100 ng/ml) (Biochem, positive control), or an appropriate dilution of *Candida* culture filtrate supernatant in the same media. The chamber was incubated for 2 h at 37 °C + 5% $CO_2$. Following incubation, the non-migrating cells on the top of the filter were removed by gently wiping the filte,r and the cells that had migrated into the bottom chamber were measured by fluorescence signal (excitation, 485 nm; emission, 530 nm) using a Tecan Spark plate reader.

### Reporting summary

Further information on research design is available in the Nature Portfolio Reporting Summary linked to this article.

## Data availability

The final cryo-EM maps have been deposited in the Electron Microscopy Data Bank under accession codes: EMD-48872 and EMD-48869. [https://www.ebi.ac.uk/emdb/EMD-48869]. The final models have been deposited in the Protein Data Bank under the accession codes: PDB-9N4D [https://doi.org/10.2210/pdb9N4D/pdb]and PDB-9N47 [https://doi.org/10.2210/pdb9N47/pdb]. The PDB entry 1EB6 was used for structural superposition analysis. The growth assay data and chemotaxis data (Figs. 4 and 5) generated in this study are provided in the Source Data file. Source data are provided with this paper.

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

## Acknowledgements

J.L.S. would like to thank Prof. Hiro Furukawa for advice, lab resources, and encouragement, as well as Dr. Dennis Thomas and Dr. Ming Wang for managing the cryo-EM and computing facilities, respectively, at Cold Spring Harbor Laboratory. J.L.S. is supported by the Institute of Biotechnology, HiLIFE, at the University of Helsinki and as an Academy Research Fellow by the Research Council of Finland (363292). D.W. is supported by the MRC Center for Medical Mycology at the University of Exeter (MR/N006364/2 and MR/V033417/1), the NIHR Exeter Biomedical Research Center (NIHR203320), and the Wellcome Trust (214317/A/18/Z). Additional work may have been undertaken by the University of Exeter Biological Services Unit. The views expressed are those of the author(s) and not necessarily those of the NIHR or the Department of Health and Social Care.Molecular graphics for several figures was performed with UCSF ChimeraX 1.7.1, developed by the Resource for Biocomputing, Visualization, and Informatics at the University of California, San Francisco, with support from National Institutes of Health R01-GM129325 and the Office of Cyber Infrastructure and Computational Biology, National Institute of Allergy and Infectious Diseases. Open access funded by Helsinki University library.

## Author contributions

A.N. designed and generated the molecular constructs and validated the fungal strains used in the growth and chemotaxis assays. E.R. and T.C. performed growth and chemotaxis assays. D.W. performed phylogenetic analyses. R.L.P. and N.S. expressed and purified the proteins. J.L.S. designed and conducted experiments involving cryo-EM and data processing. J.L.S. and R.L.P. built the molecular models. J.L.S. and D.W. wrote the manuscript. All authors reviewed the manuscript.

## Competing interests

The authors declare no competing interests.
