## [Transparent Peer review file · Nature Communications]

Structural insights into mechanisms of zinc scavenging by the *Candida albicans* zincophore Pra1

Corresponding Author: Dr Johanna Syrjänen

Version 0:

Reviewer comments:

Reviewer #1

(Remarks to the Author)

The submitted manuscript by Nore et al. is a revised version of one previously submitted to another Nature journal. It describes the cryo-EM structure of the *Candida albicans* zincophore protein Pra1, accompanied by functional assays. The manuscript is very well written, and all points raised during the previous revision process have been addressed in great detail. I now fully recommend it for publication in Nature Communications without any reservations.

Minor points:

On line 155, the first motif is referred to as the HAXXH motif, whereas on line 213 it is called the HARDH motif. To avoid confusion, the authors may wish to use a consistent name throughout the manuscript to clearly indicate that the same motif is being discussed.

Material and methods line 378: ZnSO₄ with the 4 in subscript.

Figure 1A, as well as most panels in the Extended Data Figures, appear to have a thin grey border. The authors may wish to consider removing it to enhance visual consistency.

Extended data Figure 4/7, figure legend says "Pra1 is modelled in green and deuterolysin is modelled in grey." and "Pra1 in the absence of Zn²⁺ is modelled in purple and Pra1 in complex with Zn²⁺ is modelled in green." Since the overlaid structures are experimentally determined structures and not models (generated by AlphaFold for instance), I would replace "modelled" with "shown".

Reviewer #2

(Remarks to the Author)

The authors have addressed my comments.

Reviewer #3

(Remarks to the Author)

The authors have addressed my concerns.

REVIEWERS' COMMENTS

Reviewer #1 (Remarks to the Author):

The authors have addressed my concerns.

Reviewer #2 (Remarks to the Author):

The submitted manuscript by Nore et al. is a revised version of one previously submitted to another Nature journal. It describes the cryo-EM structure of the *Candida albicans* zincophore protein Pra1, accompanied by functional assays. The manuscript is very well written, and all points raised during the previous revision process have been addressed in great detail. I now fully recommend it for publication in Nature Communications without any reservations.

Minor points:

On line 155, the first motif is referred to as the HAXXH motif, whereas on line 213 it is called the HARDH motif. To avoid confusion, the authors may wish to use a consistent name throughout the manuscript to clearly indicate that the same motif is being discussed.

We thank the reviewer for their comment and agree that this will improve clarity. We have rephrased line 155 as well as line 213 to refer to the HAXXH motif together with the residue numbers (residues 68-72).

Material and methods line 378: ZnSO₄ with the 4 in subscript.

We have corrected this.

Figure 1A, as well as most panels in the Extended Data Figures, appear to have a thin grey border. The authors may wish to consider removing it to enhance visual consistency.

Extended data Figure 4/7, figure legend says "Pra1 is modelled in green and deuterolysin is modelled in grey." and "Pra1 in the absence of Zn²⁺ is modelled in purple and Pra1 in complex with Zn²⁺ is modelled in green." Since the overlaid structures are experimentally determined structures and not models (generated by AlphaFold for instance), I would replace "modelled" with "shown".

Thank you for raising this point. We have now replaced "modelled" with "shown".

Reviewer #3 (Remarks to the Author):

The authors have addressed my comments.